# Investigating Performance in a Strenuous Physical Task from the Perspective of Self-Control

**DOI:** 10.3390/brainsci9110317

**Published:** 2019-11-09

**Authors:** Louis-Solal Giboin, Markus Gruber, Julia Schüler, Wanja Wolff

**Affiliations:** 1Sensorimotor Performance Lab, Human Performance Research Centre, University of Konstanz, 78464 Konstanz, Germany; m.gruber@uni-konstanz.de; 2Department of Sport Science, Sport Psychology, University of Konstanz, 78464 Konstanz, Germany; julia.schueler@uni-konstanz.de; 3Department of Educational Psychology, Institute of Educational Science, University of Bern, 3012 Bern, Switzerland; wanja.wolff@uni-konstanz.de

**Keywords:** Muscle fatigue, voluntary activation, self-control, performance, motivation

## Abstract

It has been proposed that one reason physical effort is perceived as costly is because of the self-control demands that are necessary to persist in a physically demanding task. The application of control has been conceptualized as a value-based decision, that hinges on an optimization of the costs of control and available reward. Here, we drew on labor supply theory to investigate the effects of an Income Compensated Wage Decrease (ICWD) on persistence in a strenuous physical task. Research has shown that an ICWD reduced the amount of self-control participants are willing to apply, and we expected this to translate to a performance decrement in a strenuous physical task. Contrary to our expectations, participants in the ICWD group outperformed the control group in terms of persistence, without incurring higher levels of muscle fatigue or ratings of perceived exertion. Improved performance was accompanied by increases in task efficiency and a lesser increase in oxygenation of the prefrontal cortex, an area of relevance for the application of self-control. These results suggest that the relationship between the regulation of physical effort and self-control is less straightforward than initially assumed: less top-down self-control might allow for more efficient execution of motor tasks, thereby allowing for improved performance. Moreover, these findings indicate that psychological manipulations can affect physical performance, not by modulating how much one is willing to deplete limited physical resources, but by altering how tasks are executed.

## 1. Introduction

Many situations require the capability to sustain physical effort for prolonged durations. Most prototypically, this is required in the context of physical exercise (e.g., running, cycling). A large body of research has been targeted towards understanding the limits to human endurance performance. Most of this research centers around the idea that a strenuous physical task is terminated because physiological limits have been reached (e.g., [1,2,3,4]). Recently, this assumption has been challenged by the idea that physical performance is limited by the perception of the effort a task induces and not by the limits of the physiological system [5,6,7,8]. This implies that psychological factors are also important regulators of how long physical effort can be sustained. More specifically, this ascribes a key role to the psychological concept of self-control in the effective regulation of physical performance. Self-control has been defined as the ‘efforts people exert to stimulate desirable responses and inhibit undesirable responses’ [9] (p. 77). For example, when a runner has the goal of breaking the two-hour mark in the marathon (given adequate physiological capabilities), self-control is needed to inhibit undesirable responses that might derail goal pursuit (e.g., slowing down because of fatigue) and to facilitate desirable responses that increase the likelihood of goal attainment (e.g., sticking to the target pace). In line with this example, it has been suggested that the decision to continue or persist in a strenuous task largely depends on how much self-control one is willing to apply [10,11,12]. 

While self-control conveys substantial societal and personal benefits [13], its exertion also carries intrinsic costs [14,15], and people prefer tasks that pose less self-control demands [16]. This begets an important question: how do individuals decide whether or not to apply control? Although a plethora of different explanations exist, many recent models conceptualize the allocation of control as some form of value-based decision [10,17]. Thus, whether or not (or how much) control is exerted hinges on a continuous cost–benefit analysis and people try to maximize the value of control [18,19]. This means that when the cost of control is expected to be higher than the expected benefits of applying it, it is no longer worth applying. Going back to the marathon example, if, at kilometer 23, one is already considerably fatigued and far behind one’s target pace, then the costs of control (i.e., dealing with all the aversive sensations that arise during the race) might outweigh their expected value (i.e., finishing the race, but not in the expected time) and one may stop racing.

The costs of applying self-control could partly explain why people do not always perform up to their physiological limit [5,7]. For example, when performing an incremental cycling exercise to exhaustion, the relevant muscles still contain a large enough functional reserve to produce the required power output. This indicates that the mechanisms inducing task failure may lie closer to central than peripheral levels [7]. These central mechanisms are most likely a mix between psychological processes (like the one hypothesized in the present study) and non-psychological processes directly altering the motor command, e.g., physiological processes inducing central fatigue (for a review, see [20]). There is now a large body of research, showing that previous exertion of self-control (on completely unrelated cognitive tasks) reduces the performance people achieve in physical tasks [12,21], or the duration for which they are able to sustain aversive sensations [22]. In the same vein, imposing demands on executive functions that rely on self-control while subjects complete a strenuous physical task also leads to performance decrements [23]. Taken together, there is now ample empirical evidence showing that performance in strenuous physical tasks depends on the application of self-control.

Research indicates that the cost of applying a given control command is monitored and computed by the dorsal anterior cingulate cortex (dACC), and then further relayed to structures like the lateral pre-frontal cortex (lPFC) that implement the control command in a top-down fashion [18]. Interestingly, during incremental cycling exercises, oxygenation of the prefrontal cortex (PFC), measured with functional near-infrared spectroscopy (fNIRS), increases with effort and decreases when subjects reach exhaustion [24], indicating that central components, possibly at the PFC level, may influence task termination decisions [8] (importantly, this drop in oxygenation is probably not task-specific, as it has also been observed in tasks that pose different physiological demands [25]). In addition, psychological strategies that supposedly automate behavior (i.e., make it less reliant on cognitive control) are associated with a reduced increase in PFC oxygenation during a strenuous physical task [26]. Thus, these findings add preliminary neuroscientific support to the conceptual and empirical evidence that emphasizes that the regulation of physical effort relies on self-control, with the PFC as the prime candidate signaling the application of self-control.

## 2. The Present Study

In this transdisciplinary study, we use labor supply theory [27,28] as a framework for investigating how changes in rewards translate to changes in psychological (perception of effort), neuronal (oxygenation in the lPFC), and physiological (neuromuscular measures of fatigue) markers of effort and performance. Labor supply theory proposes that when workers determine their preferred hours of work, they choose their subjective point of maximum utility, where labor and leisure are optimally balanced [29]. The point of maximum utility depends on the reward per hour (wage) and the number of hours one can allocate to work (budget constraint). Because it describes decision-making on the base of a balance between work and leisure (which can both be desired), the labor supply theory can allow for a description of more complex behaviors than value-based models, based only on go/no go decisions. Research has shown that the propositions of labor supply theory can be used to explain the allocation of cognitive control to a cognitive task, and that the allocation of cognitive control can be described as a utility function, that weighs the costs of control against its payoff [29]. This indicates that people treat the application of control as labor and choose to invest the amount of control that has the highest subjective utility. Such a combination can be altered by an income manipulation in ways that are predicted by labor supply theory [29]: for example, when wage per unit of time was reduced—but was compensated by an upfront payment (income compensated wage decrease; ICWD)—in a second session, subjects reduced the time spent doing demanding cognitive tasks (Study 1 in [29]). This result indicates that, even with the possibility to earn the same exact income for the same exact amount of cognitive control applied, participants have chosen not to do so (see Figure 1A). Interestingly, these results of cognitive labor/leisure decisions are in agreement with economic labor/leisure decisions, and even animal foraging-related decisions [29]. As hypothesized here and elsewhere [10], the capacity to apply cognitive and physical effort may rely on self-control. Therefore, the labor supply theory seems well suited to model the performance of a strenuous physical task within the paradigm described at the beginning of the introduction. In this paradigm, the performance of the strenuous task would depend on the chosen combination between the reward associated with the task performance and control costs that are required to continue to perform the task. 

In the present study, our goal was to assess whether the voluntary termination of a strenuous physical task could be altered by an income manipulation, in the same way as the regulation of cognitive effort applied during cognitive tasks [29]. Importantly, we have chosen a type of physical task (submaximal isometric contraction) for which it has already been shown that manipulation of self-control by ego depletion or mental fatigue experimental paradigms could affect time of disengagement (e.g., [30,31]). Therefore, we predicted that receiving an ICWD in a second session would reduce the amount of self-control participants were willing to apply and, thus, reduce the length of engagement in the strenuous physical task (Figure 1A, t1 vs. t2). Accordingly, we expected the ICWD condition to be accompanied by limited tolerance of perceived effort, less pronounced increases in lPFC oxygenation, and less severe central and peripheral muscle fatigue. 

## 3. Methods

### 3.1. Participants

The study was approved by the ethics committee of the University of Konstanz and in accordance with the Declaration of Helsinky. All methods used in this study were approved by the ethics committee of the University of Konstanz. Participants gave written informed consent before being enrolled in the study. Participants (*N* = 34; mean ± SD; age: 25.5 ± 4.7 years, height: 179.6 ± 4.8 cm, weight: 80.9 ± 11.1) took part in two experimental sessions, separated by 1 week. We recruited only male participants, since we performed electrical femoral nerve stimulations with the electrodes fixed over the gluteus muscle and the femoral triangle.

### 3.2. General Procedure

For each participant, the two experiments were always performed on the same day of the week and at the same time of day. Participants were asked to not consume alcohol or caffeine the day of the experiment and had to prevent any unusual leg exertion 48 h before the experiments. Beside this, participants had to maintain their usual activity. The first 10 participants were allocated randomly in the control or ICWD group. Then, participants were allocated in both groups to match baseline maximal voluntary contraction (MVC) and time before self-disengagement at time 1, during the strenuous physical task (a knee extension fatiguing task). Details of the anthropomorphic characteristics of each group are displayed in Table 1. During the experiment, participants were facing a computer screen that displayed the force information necessary to properly execute the given task. An experimental session proceeds as follows. First, participants were told what would happen during the experiment and gave informed consent. Then, participants were prepared for the neuromuscular and fNIRS measurements and the main task they had to perform. For this, instructions regarding the fatiguing knee extension task and how it was incentivized were given to participants by means of a standardized text, followed by standardized oral instructions (the full instruction sheet for t1 and t2 is uploaded to this manuscript as Appendix A). Before the participant started the task, we measured fNIRS baseline for 1 min. Then, the investigator said “start” and the participant started the task. Participants were kept unaware of both their group attribution and other participant’s results.

### 3.3. Experimental Manipulation and Payment

For the control group (*N* = 17) at t1 and t2, and for the ICWD group (*N* = 17) at t1, the wage was 1 Euro per minute spent on the knee extension fatiguing task. Here, the summarized instruction was as follows: “you work on the strength task as long as you want. For each trial you will receive one Euro. Thus, you will receive one Euro per minute. The task lasts until you stop, or until it is terminated because the stopping criterion [i.e., the target force is not produced anymore] has been met”. For the ICWD group, at t2, participants received an upfront payment equal to 50% of t1 income and the wage was 0.5 Euros per minute. Therefore, at t2, ICWD participants, although not explicitly made aware of this possibility, could still reach the same income–time combination (Figure 1A, crossing of budget constraint lines). Here, the summarized instruction was as follows: “you work on the strength task as long as you want. For each trial you will receive 50 Cent. Thus, you will receive 50 Cent per minute. Prior to the task you will receive a bonus of X Euro [bonus amount differed as a function of t1 performance]. The task lasts until you stop, or until it is terminated because the stopping criterion [i.e., the target force is not produced anymore] has been met”. Participants were paid 5 supplementary Euros for coming into the lab (not considered in the total income calculation). 

### 3.4. Neuromuscular Procedure

Participants started with a warm-up consisting of bodyweight exercises: 2 × 10 squats (30 s rest) followed by 2 × 3 counter movement jumps (30 s rest). Then, we taped electromyography (EMG) sensors (Trigno wireless EMG system, Delsys Inc.) on the skin, which had been previously shaved, abraded and cleaned with alcohol of the vastus lateralis (VL) and biceps femoris (BF) muscles, following SENIAM recommendation. We taped stimulation electrodes (custom made) on the end part of the gluteus maximus muscle (anode; copper, 7 × 5 cm, wrapped in a soaked sponge) and on the femoral triangle (cathode; copper, circular, 2 cm diameter, wrapped in a water-soaked sponge), in order to stimulate the femoral nerve. Participants were then seated in a custom-made chair, with their right knee forming an angle of around 100°. Participants were then tightly fastened with non-compliant straps at the torso, hip and right ankle levels. The strap at the ankle was fixed according to anatomical landmarks (2 cm higher than the line parallel to the floor passing below the lateral malleolus) to ensure an identical placement in the two experimental sessions. The ankle was fixed to a force transducer, to measure isometric knee extension force (Model 9321A, Kistler, Winterthur, Switzerland). Both EMG (high-pass- and low-pass-filtered at 20 Hz ± 10% and 450 Hz ± 10%, respectively) and force signal were sampled with a Power 1401 interface (Cambridge Electronic Design, Cambridge, UK) at 4000 Hz and stored on a computer with the Signal software (Cambridge Electronic Design). Participants had their arms crossed over their chest during the measurement parts of the experiment.

We then stimulated (squared pulse of 1 ms duration, DS7A stimulator, Digitimer) the femoral nerve with incremental stimulation intensities, until reaching the maximal amplitude of the M-wave (Mmax) in the VL muscle and until the twitch of knee extensors muscles when at rest did not increase anymore. Intensity was then set at 150%. Then, participants performed 15–20 incremental voluntary contractions, until reaching around 90% of perceived maximal effort. These contractions had two purposes: to warm-up, specifically to produce an isometric MVC, and to familiarize participants with the control of the cursor. All participants were able to produce a stable 15% MVC contraction and an MVC with a decent force plateau. Then, after 2 min rest, participants performed an MVC for 3 s. For every MVC performed during the experiment, participants were encouraged with standardized shouts. During this MVC, peripheral nerve stimulation (PNS) was delivered during the contraction plateau (to obtain the amplitude of the superimposed twitch) and at rest (to obtain the amplitude of the potentiated twitch at rest, Ptw), 2 s after the end of the contraction. This procedure allows the estimation of voluntary activation (VA) using the interpolated twitch method [32] with the following formula: VA = (1–superimposed twitch amplitude/potentiated twitch amplitude) × 100. This procedure was done a second time after 2 min of rest. If an increase of 5% or more in MVC was observed at baseline, the procedure was repeated.

For the readers not familiar with neuromuscular measurements, a decrease in MVC following the fatigue task indicates the occurrence of neuromuscular fatigue. Roughly, neuromuscular fatigue can stem from events occurring in the nervous system (central fatigue) and events occurring distal to the neuromuscular junction (peripheral fatigue). A decrease in VA indicates a decrease in the capacity to voluntarily recruit muscles and is an estimate of central fatigue. A reduction in Ptw indicates a reduction in muscle excitability and is an estimate of peripheral fatigue. The present measurements can therefore help to quantify the amount of physiological resources spent during the task. For more details, please see [33].

### 3.5. Knee Extension Task

The knee extension task consisted of keeping a cursor over a threshold line (dashed line in Figure 2) displayed on a screen in front of the participant until self-disengagement (1 m distance). If, at one time point, the cursor stayed below the line for more than 2 s, the task was terminated. The cursor was moved by the isometric knee extension contraction, and the threshold was equal to 15% of their maximal voluntary contraction (MVC). Participants were naïve regarding this calculation. The task was displayed on the screen in 1 min long frames. Before starting the task, standardized oral instructions were given: “You can stop at any time you want. When you stop, say “I stop” and then relax totally your leg and then immediately contract maximally”. At the beginning of the frame, an investigator indicated that the participant could initiate the contraction by saying “start”. After 4 s, the investigator told the participant how much money he had already earned and how much money he will earn if he completes the 1 min frame (e.g. “you have earned 1 euro, if you finish this frame you will earn 2 euros”). After 30 s, a peripheral nerve stimulation (PNS) of the femoral nerve was triggered to elicit a maximal M-wave. After 48 s, participants were asked to give their rate of perceived exertion (RPE, scale of 1 to 10, with the possibility of going beyond 10 [34]). At 57 s, the participant was told to fully relax their knee extensor muscles (“relax”). This rest had two purposes. First, it allowed us to ensure that there was no drift in the force transducer during the task. Second, this short rest allowed the muscle to recover substantially (as seen with the recovery of the force production variation at the beginning of the next frame [35]), therefore making the task’s termination “more psychological”. The participant had to say “I stop” when stopping the task, and fully relax their leg. As soon as the cursor reached zero on the y axis and, following the instructions of the investigator (“fully relax the leg, maximal contraction, go!”), the participant had to produce an MVC, and PNS was performed to measure VA and Ptw. Strong verbal encouragements were given. Other than this, there was no interaction between participants and investigators. 

### 3.6. fNIRS

Changes in cerebral Oxyhemoglobin were continuously measured with an 8 Emitter + 8 Detector multichannel continuous-wave fNIRS imaging system (NIRSport, NIRx Medical Technologies LLC, NY, USA). The NIR wavelengths were 760 nm and 850 nm, and data were collected at a sampling rate of 7.81 Hz. Two 4 emitter + 4 detector arrays were bilaterally positioned over scalp sites that corresponded to the lPFC (see Figure 3A). Emitters and detectors were positioned according to the international 5/10 system: E1 at F1, E2 at AF3, E3 at FC3, E4 at F5, D1 at F3, D2 at AF7, D3 at FC5, D4 at F7, E5 at F6, E6 at AF4, E7 at FC4, E8 at F2, D5 at F8, D6 at AF8, D7 at FC6, and D8 at F4. This montage was designed to measure activity over the dorsal (Emitter–Detector combinations: E1_D1, E2_D1, E3_D1, E6_D8, E7_D8, E8_D8, E2_D2, E3_D3, E6_D6) and ventral (Emitter–Detector combinations: E4_D1, E4_D2, E4_D3, E5_D5, E5_D6, E5_D8) areas of the lPFC. Channels of interest were emitter–detector pairs with 3 mm separation. This resulted in nine channels on the left (channels 1–9) and nine channels on the right (channels 10–18) hemisphere. Channels 9, 12, and 16 had to be excluded from the analyses, due to detector malfunction. The probes were fixated in a NIRScap (EASYCAP GmbH, Herrsching, Germany) with an interoptode distance of 30 mm. The NIRScaps for optode placement were available in three different sizes (head circumferences of 54, 56, and 58 cm) and suitable for all subjects. To ensure better signal quality, a retaining overcap (EASYCAP GmbH, Herrsching, Germany) was placed over the NIRScap. 

### 3.7. Analysis and Statistics

fNIRS data of each subject were preprocessed using HOMER2 (MathWorks Inc., 2016) [36]. The enPruneChannels function was used to remove channels when the signal was too weak or too strong and then optical intensity was converted to optical density using the Intensity_to_OD function. Then, a discrete wavelet transform was performed to identify and correct motion artifacts [37]. Finally, data were low-pass filtered (0.5 Hz) and converted to Oxy- and Deoxyhemoglobin with the modified Beer–Lambert law [38], using the default differential path length factor of 6.0 for both wavelengths. Finally, fNIRS data were time-normalized, to allow for comparison between groups and sessions across the whole duration of the task. 

We measured the peak-to-peak amplitude of MVC and Ptw, and calculated VA with the amplitude of the superimposed twitch and Ptw. The baseline values were taken from the biggest MVC performed before the fatiguing task. For the EMG, we used root mean square over the whole frame (from 3 s to 28 s and from 32 s to 57 s, to avoid analyzing the contraction initiation and termination, as well as the Mmax potential) and normalized it to the amplitude of the Mmax measured during the same frame. The RPE and EMG were time-normalized, in order to make comparisons between sessions and groups across the duration of the task. For the mean force produced during the task, we averaged the force for the whole duration of the task (without the final MVC). Statistics were performed with JASP (for the two-way ANOVAs) and with R (for the linear mixed-effects analyses). We used two-way ANOVAs with group (control vs. ICWD, between subject) and session (t1 vs. t2, within subject) as a factor in the time before self-disengagement in the knee extension task, income, and mean force during the task. These ANOVAs were followed by paired t-tests at the time factor level. Three-way ANOVAs were used to test whether groups, session, or time within sessions (pre fatigue task vs. post fatigue task, within subject) had an effect on MVC, VA and Ptw. We estimated linear mixed-effects models (LMM) with LME4, and used the Satterthwaite approximation for degrees of freedom implemented in LMERTEST to establish the significance of fixed-effects. All LMM were estimated with random effects for participants. We tested whether there was an interaction between groups and time within sessions separately for the RPE and EMG at t1 and t2. In the model estimate for fNIRS data, we tested the interaction between groups and session and added a fixed effect of the time within a session.

## 4. Results

### 4.1. Behavioral Results

ANOVAs showed an effect of the sessions (F_1,32_=13.6, *p* < 0.001; F_1,32_ = 9.3, *p* = 0.005) but no clear sessions × group interaction (F_1,32_ = 3.23, *p* = 0.08; F_1,32_ = 0.31, *p* = 0.57) between the time before self-disengagement and income, respectively, during the knee extension task. In the control group, no change in time before self-disengagement and the reward obtained from t1 to t2 was observed (Figure 1B; t_16_ = −1.4, *p* = 0.17 for both performance and income), indicating the robustness of the initially determined point of maximum utility. In the ICWD group, income and time before self-disengagement increased at t2 (Figure 1C; t_16_ = −3.7, *p* = 0.002 for both). 

### 4.2. Perceived and Physiological Effort

LMM showed no difference in the time-normalized rate of perceived exertion (RPE), measured every minute during the task, between groups for each experimental session (see Figure 4). The Three way ANOVAs showed no difference between groups or sessions in maximal voluntary contractions (MVC), voluntary activation (VA) or the potentiated twitch at rest (Ptw). Only an effect within sessions (pre vs. post strenuous task) was observed for each dependent variable (*p*-values all < 0.001; see Table 2 for numerical values). With LMM, we observed no difference between groups and across the task for each experimental session in vastus lateralis (VL) electromyogram (EMG), normalized to Mmax (see Figure 5). Finally, LMM revealed a significant increase of oxygenation over the course of the time-normalized fatiguing task (F_9,1116.2_ = 72.91, *p* < 2.2e–16). In addition, there was an interaction between experimental session and group (F_1,67.69_ = 10.99, *p* = 0.0014), indicating higher oxygenation in the control group in the second experimental session (see Figure 3). 

### 4.3. Task Efficiency

A two-way ANOVA showed the session × group interaction effect of the total mean force produced during the knee extension task (F_1,32_ = 4.95, *p* = 0.033; see Figure 6), indicating a lower mean force produced during the task at t2 for the ICWD group. 

## 5. Discussion

This is the first study to use labor supply theory to explain self-controlled persistence in a physically demanding task. We found that receiving an ICWD in a second session resulted in longer time before disengagement in a strenuous physical task, and not the expected decrease in performance. This is a puzzling result: from a utility maximization perspective this increase appears to be irrational, because, compared to t1, each unit of income increase requires two units of effort. Thus, the ICWD manipulation apparently triggered a shift in the subjective point of maximum utility, leading subjects to apply more self-control, allowing them to tolerate greater physical exertion, in order to increase income. However, if heightened motivation is the reason for the improved performance, then this should have been accompanied by a greater depletion of psychological (RPE), neuronal (lPFC oxygenation), and physiological (neuromuscular measures of fatigue) markers of effort and performance. To test this, we estimated the task-induced physiological costs by measuring muscle fatigue and it’s central and peripheral components. Further, we estimated the psychological costs by assessing RPE throughout the task, and we indirectly estimated the application of self-control by continuously monitoring blood oxygenation in the lPFC throughout the task [15,39,40]. Interestingly, for none of the physiological measures did we observe greater levels of fatigue in the ICWD group, indicating that the longer time before disengagement was not achieved via a greater depletion of physiological resources. This finding was mirrored by the observation that participants in the ICWD group did not report higher ratings of perceived exertion. Thus, the longer time before disengagement was not achieved by a greater tolerance to the sensation of effort. This result is important, since previous research indicates that perceived effort (as measured with RPE) is “the cardinal exercise stopper” [41], thereby making tolerance of effort a crucial determinant of time before disengagement. Our results are in line with this reasoning, as participants apparently did not go over a certain threshold of RPE, irrespective of incentivization. Finally, compared to the control group, participants in the ICWD group displayed a less pronounced increase in lPFC oxygenation over the course of the task, indicating comparatively less activation in an area that is relevant for the application of control. Taken together, the longer time before disengagement was achieved without greater depletion of psychological and physiological resources and even with reduced involvement of self-control relevant brain areas. As no differences in the amount of resource depletion could be measured, the longer time before disengagement likely stems from a change in the utilization of available resources. Indeed, post-hoc analyses revealed that the mean force produced during the task was reduced in the ICWD condition. This means that subjects were able to produce force curves that were closer to the target force of t2. This equates to a more efficient use of resources while performing the task, allowing for a longer time before self-disengagement without a change in physiological and psychological costs. 

## 6. Implications

Our findings are surprising in at least two ways: first, in contrast with our hypotheses, participants in the ICWD condition increased their task duration at t2. Second, this increase was not achieved by a greater depletion of physiological and psychological resources but by a more efficient use of available resources and a less pronounced engagement of a cerebral correlate of self-control application. While we did not expect to find this pattern of results, we believe that research on the effects of rewards on movement efficiency and the predictions of labor supply theory might allow for a tentative explanation of these findings: the increase in physiological movement efficiency that was observed in the ICWD condition falls in line with research demonstrating that the noise or metabolic costs of low-demand physical tasks can be influenced by rewards [42,43]. Thus, performing a low-demand physical task with greater accuracy (i.e., with less noise in the movement) requires the allocation of more self-control [42]. However, the control–noise relationship is likely reversed in strenuous physical tasks: if a task is performed more efficiently (i.e., with less noise), less self-control is needed to fight against the urge to disengage from the task. In other words, if a physically demanding task can be performed more efficiently, the amount and magnitude of aversive signals (e.g., muscle receptor feedback [44], corollary discharge [45]) that have to be integrated by areas relevant for self-controlled task continuation should be reduced. In line with this, a steeper increase in lPFC activation was observed in the control group from time 1 to 2, but no such change occurred in the ICWD group. Thus, if the ICWD manipulation lowered the amount of self-control participants were willing to allocate to the task (as measured by lPFC oxygenation), then—ironically—this might have, in fact, increased the time before disengagement, by allowing for a more efficient task execution. To summarize, looking solely at performance, our results seem to suggest that the predictions derived from labor supply theory do not hold in the case of regulation of physical effort. However, it is important to note that lPFC oxygenation, which was used to operationalize the allocation of self-control, changed in a way that was consistent with the expected effect of the ICWD manipulation. Thus, our results suggest that the amount of self-control one applies is not linearly—and is possibly even counterintuitively—related to the to-be regulated physical performance. In our data, this is exemplified by a counterintuitive increase in performance, despite a less pronounced involvement of the lPFC. Thus, further research should focus on disentangling the relationship between self-control and physical performance, and specifically investigate instances where less control leads to a more efficient use of resources and, hence, improved performance.

According to research on the neuronal mechanisms that govern control [18,19], the shift in movement and cognitive control is monitored and specified by the dACC. One possible explanation for our findings is that the dACC diverted the control-signal from a mainly top-down regulated control signal (i.e. lPFC activity) to striatal areas that play a role in movement control. Indeed, these latter networks are sensitive to reward and could modulate movement efficiency [46]. This falls in line with the dopaminergic-dependent, reward-induced reduction in noise demonstrated in low-demand physical tasks [19,42]. Supporting this, compared to healthy controls, neurological patients who suffer from disease-induced decrements in motor performance experience steeper increases in RPE [47] and higher activity in the cortico-striatal network, which has been implicated in fatigue [48]. This has been interpreted as a compensatory activation to account for the inordinate amount of resources (i.e., cognitive effort) that patients are required to spend to reach an intended outcome (physical task). 

## 7. Limitations

It must be noted that the task structure used in the present study differs in important ways from the only other study we know of that used labor supply to predict the allocation of self-control controls [29]. In our study, subjects performed a physical task, whereas, in the study by Kool and Botvinick [29], subjects performed a cognitive task. Thus, the latter is a more direct measure of control allocation, whereas, in our study, self-control allocation represents a mediator between subjects’ capacity to perform the task and actual task performance [10]. The tasks also differed on a structural level: instead of a task with the possibility of allocating time spent doing a difficult or an easy contraction, according to participant preference (which would mirror Kool and Botvinick’s approach [29]), participants had to perform a continuous contraction until self-disengagement. We opted for this setup for two reasons. First, physiological processes leading to neuromuscular fatigue are dependent on the way contractions are performed and the resting time between hard efforts (for example, see [49]). If using a paradigm with choices of physical effort allocation, we would not be able to control whether a different combination of the total time spent doing a hard contraction and subsequent income was induced by a better strategy, limiting the accumulation of neuromuscular fatigue (e.g., a more optimal effort/rest sequence) or due to the psychological manipulation. A scripted set of contractions, or a continuous contraction, like in the present study, reduces this bias and allows the quantification of the effect from the psychological manipulation on physical effort. Second, many real life decisions are about whether or not to further invest effort into one, ongoing task and researchers have explicitly called for the investigation of such scenarios within the framework of labor supply theory [29]. Thus, we chose to respond to this call and use a single continuous task. 

It should also be noted that, in the lPFC activity results, while we expected to see a lower slope in activation increase in the ICWD group compared to the control group, it is surprising that this difference is driven by an altered oxygenation pattern in the control group and not in the ICWD group. 

Finally, evidence indicates that manipulation of self-control by the means of ego depletion or a mental fatigue experimental paradigm affects whole-body and isolation tasks differently [21]. Therefore, the present results may not generalize to all kinds of strenuous physical effort.

## 8. Conclusions

We tested the hypothesis that a manipulation intended to reduce the amount of self-control participants were willing to apply to a strenuous physical task would also reduce performance of an isometric knee extension task. While such an ICWD has been shown to reduce the amount of effort participants were willing to invest into a cognitive task [29], on a phenomenological level a reversed effect was observed in regards to a physical task: surprisingly, reducing the rewards participants could obtain per unit of time led to better performance in the strenuous physical task. This increase was not achieved with an increase in psychological (RPE), neuronal (lPFC oxygenation), or physiological costs (neuromuscular measures of fatigue) but, rather, with an increase in movement efficiency. Thus, on the neuronal level, our results appear to be in line with labor supply theory, but the mechanisms by which control affects net outcome in a strenuous physical task (where subjects have to make do with limited resources) might be less straightforward and actually cause a behavioral outcome that is in contrast to the theoretical predictions. This finding highlights the intriguing possibility that, in some situations, higher persistence in demanding physical tasks may not necessarily require more self-control. In such a situation, a shift in motivation may not lead to a more excessive use of resources, but to more efficient resource utilization. This result may have important implications for the effects of reward structures in sport, economic, psychological and motor control domains.

## Figures and Tables

**Figure 1 brainsci-09-00317-f001:**
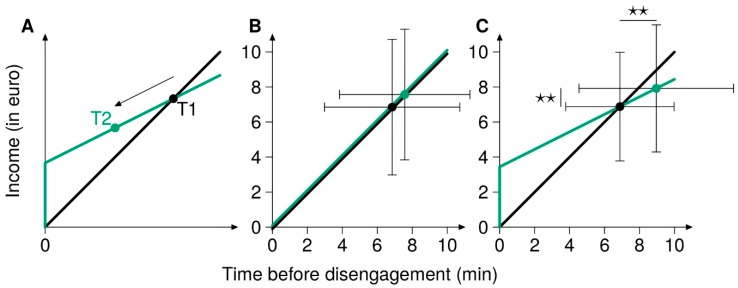
The utility of cognitive control. In this figure, adapted from Kool and Botvinik [29], the black and green dots correspond to the mean combination of income (in euro) and time (in min) before disengagement at t1 and t2, respectively. The black and green lines represent the mean budget constraint at t1 and t2, respectively. The budget constraint line corresponds to all the possible combinations of income and time before disengagement, in the given experimental context. The higher ordinate at the abscissa zero corresponds to the mean upfront payment. The error bars represent SD and the double five-pointed stars correspond to a *p*-value < 0.01. (**A**) Prediction of the iso–utility curve of the task with an ICWD condition at t2, according to the model of cognitive effort control [29]. (**B**) Results at t1 and t2 of the control group. (**C**) Results at t1 and t2 of the ICWD group.

**Figure 2 brainsci-09-00317-f002:**
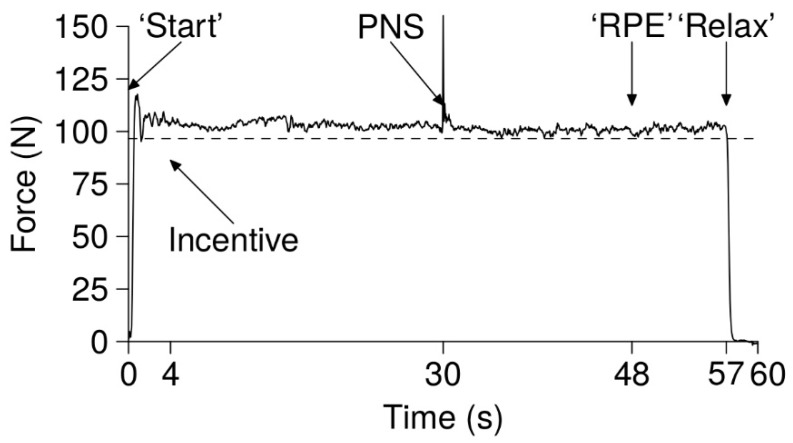
The knee extension task. The task consisted of keeping a cursor over a threshold line (dashed line on the present figure, corresponding to 15% maximal voluntary contraction(MVC)) displayed on a screen in front of the participant, until self-disengagement, by means of an isometric knee extension. If, at one time point, the cursor stayed below the line for more than 2 s, the task was terminated. Participants were told that they could stop the task at any time they wanted. The task was displayed on the screen in 1 min long frames. At the beginning of the frame, an investigator indicated that the participant could initiate the contraction by saying “start”. At 4 s, the investigator told the participant how much money he had already earned and how much money he would earn if he completes the 1 min frame. At 30 s, a peripheral nerve stimulation (PNS) of the femoral nerve was triggered to elicit a maximal M-wave. At 48 s, participants were asked to give their rate of perceived exertion. At 57 s, the participant was told to fully relax their knee extensor muscles.

**Figure 3 brainsci-09-00317-f003:**
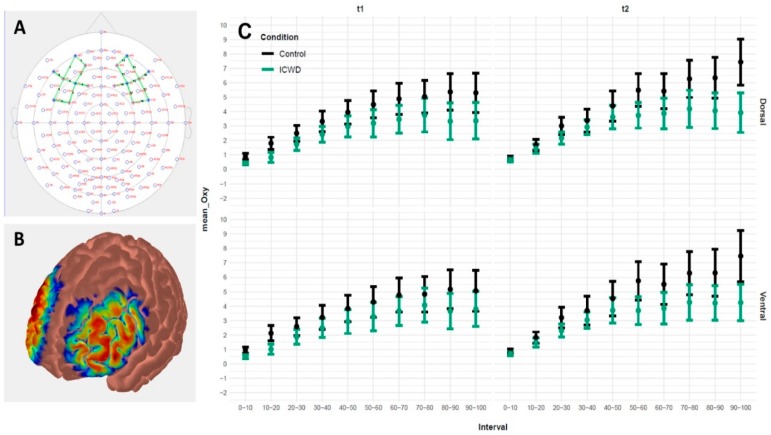
fNIRS measurements. (**A**). The sensitivity profile was created with Atlas Viewer (Aasted, et al., 2015) and it indicates that the chosen optode placements capture the lateral prefrontal cortex (lPFC) reasonably well. It represents Monte Carlo random walks of 1e7 photons (per optode) migrating through a standard atlas (Colin27, **B**). Panel **C** displays the change in oxygenation throughout the fatiguing task [0%–10%] vs. [10%–20%] vs. [90%–100%]) and is displayed separately for experimental conditions, experimental session and regions of interest (ROI; ventral and dorsal lPFC). Error bars represent SEM. ICWD corresponds to income compensated wage decrease.

**Figure 4 brainsci-09-00317-f004:**
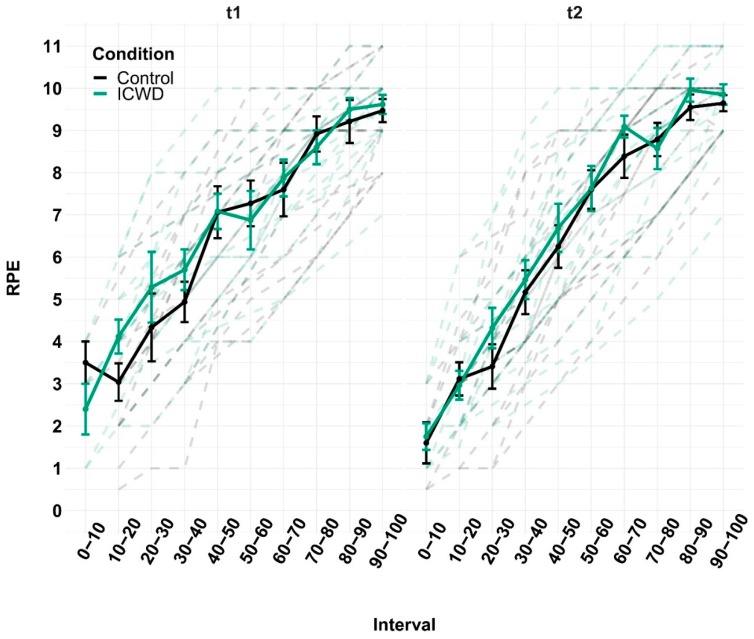
Rate of perceived effort. Mean rate of perceived exertion (RPE) are displayed throughout the time normalized task (Interval) during the first (t1) and second (t2) experimental session for the control (black) and ICWD (green) groups. The dashed lines correspond to individual data. Error bars correspond to SEM.

**Figure 5 brainsci-09-00317-f005:**
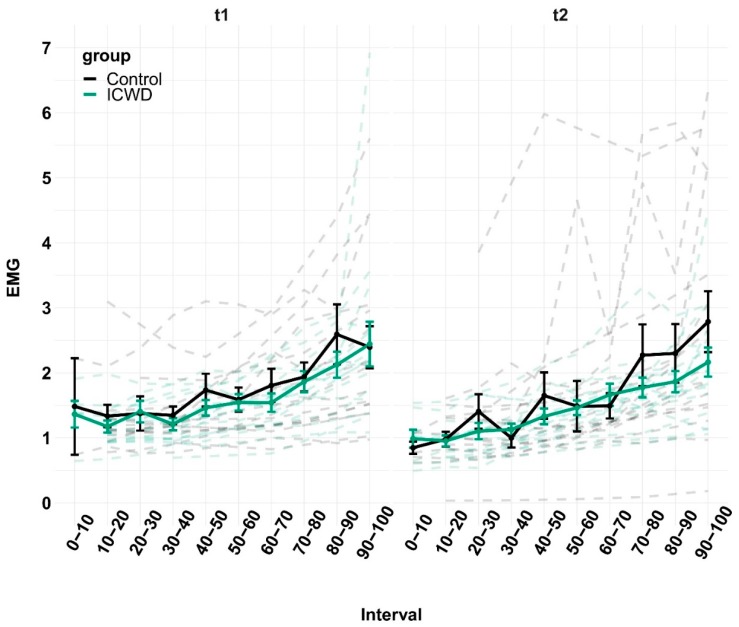
EMG. Mean rms EMG, normalized to Mmax (%), are displayed throughout the time normalized task (Interval) during the first (t1) and second (t2) experimental session for the control (black) and ICWD (green) groups. The dashed lines correspond to individual data. Error bars correspond to SEM.

**Figure 6 brainsci-09-00317-f006:**
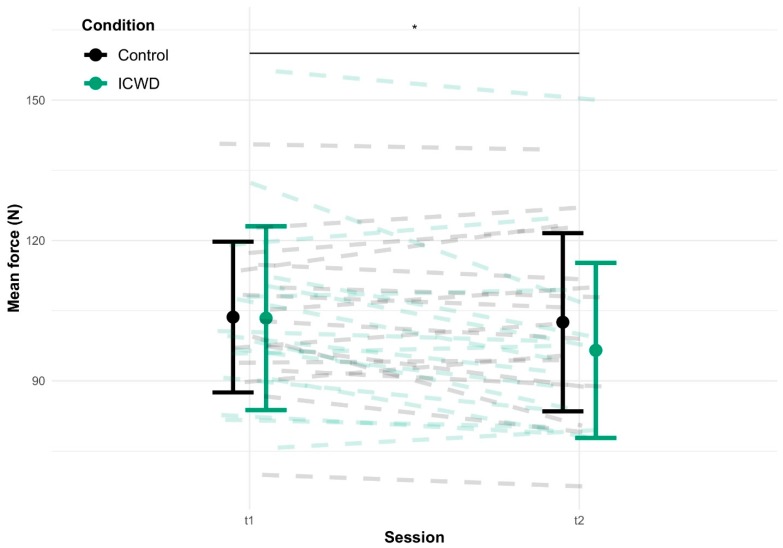
Task efficiency. Display of the mean force (in N) exerted during the whole task duration at t1 and t2 for both groups. The dashed lines represent individual data and the error bars represent SD. The black horizontal bar crowned by a star represents a time × group interaction with a p-value = 0.03.

**Table 1 brainsci-09-00317-t001:** Anthropometric characteristics of each group. Height in cm, weight in kg and age in years. Mean ± standard deviation. For each variable, the difference between groups was assessed with a t-test. ICWD corresponds to income compensated wage decrease.

	Control (*N* = 17)	ICWD (*N* = 17)	*p*-Value
Height	179.1 ± 3.3	180.2 ± 5.9	0.49
Weight	81.5 ± 8.9	80.4 ± 13.2	0.78
Age	25.7 ± 3.8	25.2 ± 5.7	0.78

**Table 2 brainsci-09-00317-t002:** Neuromuscular results. Mean ± SD of maximal voluntary contraction (MVC) in N), voluntary activation (VA in %) and potentiated twitch at rest (Ptw in N) of both groups (Control vs. ICWD) at T1 and T2 sessions and pre- and post-knee extension task.

Group	Control				ICWD			
Session	T1		T2		T1		T2	
Within Session	Pre	Post	Pre	Post	Pre	Post	Pre	Post
MVC (N)	651 ± 131	354 ± 115	660 ± 141	367 ± 123	637 ± 121	360 ± 179	628 ± 123	376 ± 163
VA (%)	93.1 ± 5.3	81.8 ± 15	92.6 ± 6.3	84.5 ± 17.3	94.4 ± 5.5	84.1 ± 11.1	93.7 ± 3.9	86.6 ± 13
Ptw (N)	153 ± 29	74 ± 27	154 ± 34	72 ± 3	150 ± 21	76 ± 41	146 ± 24	81 ± 35

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
