# Peer review of "Investigating Performance in a Strenuous Physical Task from the Perspective of Self-Control"

_brainsci, 2019, doi:10.3390/brainsci9110317_

Round 1

Reviewer 1 Report

The authors present an experimental study examining the effect of two different income procedures (Income compensated wage decrease vs. stable income) on persistence in a strenuous knee extension isometric contraction task. The authors use the labor/leisure tradeoff approach developed by Kool and Botvinick (2014) as theoretical framework. The experiment is well conducted and the methodology appropriate to test the hypothesis. The authors did not replicate Kool and Botvinick results. Some points need to be addressed to improve the paper.

1. Introduction, page 2, lines 66-69: In the example of the marathon, I propose to the authors the replace the verb ‘to terminate’ with ‘to stop’, because I think that when costs are higher than benefits the athlete will stop the race rather than terminate the race.

2. Methods, page 3, lines 129-133: Please provide a table with main characteristics (age, MVC at T1, BMI, number of subjects per group, number of males and females per group) for both groups separately and compare group characteristics with t-test.

3. Methods, page 4, lines 145-146: Please provide the instructions given to the participants by the mean of the standardized text, particularly concerning the motor task (two-three main sentences concerning the persistence instructions).

4. Methods, page 4, lines 181-185: Please explain simply for the reader what is the main interest to measure: (1) the amplitude of the superimposed twitch, (2) the amplitude of the potential twitch at rest and (3) the voluntary activation (VA).

5. Methods, page 5, lines 210-216: The four sentences describing the payment procedure must be given earlier in the ‘Methods’ section. I suggest to give this information in the ‘General procedure’ sub-section.

6. Discussion, page 7, lines 309-310: The authors claim “longer time to disengagement was not achieved by a greater tolerance to the sensation of effort”. Do they know references showing a relationship between tolerance of effort and time to disengagement? If the answer is affirmative, I invite the authors to cite these references.

7. Discussion: The authors showed that the participants in the ICWD condition increased their time to disengagement with a more efficient use of available resources. This result suggests that the participant of the ICWD group used a less effortful strategy to perform the task and this result is in agreement with the Hockey’s state regulation model of compensatory control (2017).

Author Response

Reviewer 1

The authors present an experimental study examining the effect of two different income procedures (Income compensated wage decrease vs. stable income) on persistence in a strenuous knee extension isometric contraction task. The authors use the labor/leisure tradeoff approach developed by Kool and Botvinick (2014) as theoretical framework. The experiment is well conducted and the methodology appropriate to test the hypothesis. The authors did not replicate Kool and Botvinick results. Some points need to be addressed to improve the paper.

### We thank you for your time and work. We have tried to answer your comments as best as possible, and we believe that these changes helped to make the manuscript better.

Introduction, page 2, lines 66-69: In the example of the marathon, I propose to the authors the replace the verb ‘to terminate’ with ‘to stop’, because I think that when costs are higher than benefits the athlete will stop the race rather than terminate the race.

### We agree with you and have replaced “terminates” by “stops”.

Methods, page 3, lines 129-133: Please provide a table with main characteristics (age, MVC at T1, BMI, number of subjects per group, number of males and females per group) for both groups separately and compare group characteristics with t-test.

### We have added a Table (Table 1) with these variables and a t-test. The table us mentioned in section 3.2 (General procedure). It must be noted that we have recruited only males since we used femoral nerve electrical stimulations (electrodes positioned over the gluteus muscle and the femoral triangle) and since investigators were male. This detail was initially described in section 3.3 (neuromuscular procedure), but we moved it in section 3.1 (Participants) to make it more visible.

Methods, page 4, lines 145-146: Please provide the instructions given to the participants by the mean of the standardized text, particularly concerning the motor task (two-three main sentences concerning the persistence instructions).

### We have added details and we have now included the instructions as a supplementary material.

Methods, page 4, lines 181-185: Please explain simply for the reader what is the main interest to measure: (1) the amplitude of the superimposed twitch, (2) the amplitude of the potential twitch at rest and (3) the voluntary activation (VA).

### This is a good point! We have added the following paragraph:

“For the readers not familiar with neuromuscular measurements, a decrease in MVC following the fatigue task indicates the occurrence of neuromuscular fatigue. Roughly, neuromuscular fatigue can stem from events occurring in the nervous system (central fatigue) and events occurring distal to the neuromuscular junction (peripheral fatigue). A decrease in VA indicates a decrease in the capacity to voluntarily recruit muscles and is an estimate of central fatigue. A reduction in Ptw indicates a reduction in muscle excitability and is an estimate of peripheral fatigue. The present measurements can therefore help to quantify the amount of physiological resources spent during the task. For more details, please see [26].”

Methods, page 5, lines 210-216: The four sentences describing the payment procedure must be given earlier in the ‘Methods’ section. I suggest to give this information in the ‘General procedure’ sub-section.

### We agree with the reviewer and have moved this information in the “General procedure” section.

Discussion, page 7, lines 309-310: The authors claim “longer time to disengagement was not achieved by a greater tolerance to the sensation of effort”. Do they know references showing a relationship between tolerance of effort and time to disengagement? If the answer is affirmative, I invite the authors to cite these references.

### We thank you for this remark. We took this opportunity to add details and a reference:

“This result is important, since previous research indicates that perceived effort (as measured with RPE) is “the cardinal exercise stopper”, thereby making tolerance of effort a crucial determinant of time to disengagement [34].” (Staiano et al., 2018, The cardinal exercise stopper: Muscle fatigue, muscle pain or perception of effort?)

Discussion: The authors showed that the participants in the ICWD condition increased their time to disengagement with a more efficient use of available resources. This result suggests that the participant of the ICWD group used a less effortful strategy to perform the task and this result is in agreement with the Hockey’s state regulation model of compensatory control (2017).

### We thank you for this suggestion. However, we could only find the following reference:

Hockey, 1997. Compensatory control in the regulation of human performance under stress and high workload: A cognitive-energetical framework. Biological Psychology.

We agree that our results share very interesting similarities with Hockey’s model. However, we think that our results do not completely fit within this model. The present task seems to correspond to the situation “ when the perceived level of difficulty is too great to be met by small adjustments to the working effort budget.” (p.9, Hockey 1997). In this situation, the regulatory model proposes 2 options to resolve the problem of a “discrepancy between increasing demands and the upper point for effort expenditure”. The first option is the strain coping mode, where the task performance can be maintained, but at the expense of more energetical resource. The second option is the passive coping mode, where there is a downward adjustment of performance target (e.g. reduction of the target level of accuracy or speed). In the present manuscript, performance seems to be increased while the energetic cost (as we have measured it) doesn’t seem to increase. Therefore, we agree that the participants of the ICWD group may have used a less effortful strategy to perform the task, but not in the way of proposed by the regulatory model: we have neither an increase in energetic cost nor a decrease in performance target (actually the contrary for the latter point). Nonetheless, we recognize that the accrued accuracy observed in the present study (which is a more efficient use of resource) could correspond to the “strategic adjustment” described by Hockey. Furthermore, it could be that due to the inherent limitation of our measurements, the accrued performance was the result of a reconfiguration of other available resource (in agreement with the model), which we have not measured or cannot estimate. Due to the only partial fit to the model of Hockey, and if the reviewer agrees, we would prefer not to refer to it as it may increase the complexity of the discussion.

Reviewer 2 Report

The topic of this article is very interesting and I enjoyed reading it! Key strengths of the paper include the novelty of the research question, the theoretical underpinning, as well as the methodological protocol. I just have a few minor comments, which are discussed below:

Broad Comments:

In the introduction section, you provided a good overview of self-control. I also like the sporting example; this helped to explain self-control further. You referred to some relevant research that has explored the effects of self-control on physical performance, as well as highlighted the potential mechanisms for this effect. My only comment here is it may be beneficial to include some research that has explored the effects of self-control on isolated muscle performance, as the performance measure in your study was exploring an isometric knee extension task. For instance, see Bray et al., 2008; 2011, although they did not use a knee extension task, they examined handgrip performance. Although your results were not as you expected, I think you have provided some good explanations as to why this may be the case. I just have a quick question; do you think the type of performance task may have potentially been a reason as to why you did not see the results you expected? For instance, many of the physical tasks that have been used in previous research are endurance-based tasks (e.g., 10km cycling time trial). It could be argued that these require high levels of self-control (e.g., for pacing, to resist the feelings of pain/discomfort etc.). It may be beneficial to discuss this in your discussion section.

Specific Comments:

In your methods section, on line 254 and 257 (page 6) you state 2 ways ANOVA’s, I think this would be better written as 2 way ANOVA’s. This is also the case when you refer to ANOVA’s in your results section.

Author Response

Reviewer 2

The topic of this article is very interesting and I enjoyed reading it! Key strengths of the paper include the novelty of the research question, the theoretical underpinning, as well as the methodological protocol. I just have a few minor comments, which are discussed below:

### We thank you very much for your time and effort spent on this manuscript. We also thank you for your appreciation of the manuscript.

Broad Comments:

In the introduction section, you provided a good overview of self-control. I also like the sporting example; this helped to explain self-control further. You referred to some relevant research that has explored the effects of self-control on physical performance, as well as highlighted the potential mechanisms for this effect. My only comment here is it may be beneficial to include some research that has explored the effects of self-control on isolated muscle performance, as the performance measure in your study was exploring an isometric knee extension task. For instance, see Bray et al., 2008; 2011, although they did not use a knee extension task, they examined handgrip performance. Although your results were not as you expected, I think you have provided some good explanations as to why this may be the case. I just have a quick question; do you think the type of performance task may have potentially been a reason as to why you did not see the results you expected? For instance, many of the physical tasks that have been used in previous research are endurance-based tasks (e.g., 10km cycling time trial). It could be argued that these require high levels of self-control (e.g., for pacing, to resist the feelings of pain/discomfort etc.). It may be beneficial to discuss this in your discussion section.

 ### We thank you for your suggestions and for your comment. We have changed the introduction and tried to integrate more of the literature you refer to. We also agree with you that the present effect is possibly task-specific, since we suppose that interactions between self-control and other mechanisms regulating physical performance are task-specific as well. Supporting this hypothesis, we have recently published a meta-analysis showing that the effect of ego-depletion or mental fatigue on a subsequent physical task is lower when the physical task is a “full body” task (e.g. running or cycling) compared to when the physical task is an “isolation” task (e.g. handgrip, or a knee extension) (https://psyarxiv.com/mr5pk/, in press). We have added this point in the limitation section.

Specific Comments:

In your methods section, on line 254 and 257 (page 6) you state 2 ways ANOVA’s, I think this would be better written as 2 way ANOVA’s. This is also the case when you refer to ANOVA’s in your results section.

### We thank you for noticing that! We have made the corrections.

Reviewer 3 Report

Review of “Investigating performance in a strenuous physical task from the perspective of self-control” Manuscript Brain Sciences

General comments:

The aim of this study was to test the effect of income compensated wage decrease (considered as ego-depletion task) on persistence in a strenuous physical task. The authors have used the labor supply theory in order to make the hypothesis that self-control could regulate voluntary strenuous physical task. Contrary to prediction, the authors didn’t validate their hypothesis and some explanations are given.

Despite interesting purpose, four main criticisms can be made:

First. the authors wrote that “the capability to endure physical effort is an important feature for species survival”. Can they give some examples to illustrate this point of view? One can be in desagreement with that. The first paragraph of the introduction section is very confusing and must be rewritten. The authors mix several models (cost-benefits, resource) and the understanding becomes complicated. Paragraph 2 is redundant with the first paragraph and deserves clarifications. The authors have to focus on one model and sufficiently describe it to propose clear hypotheses. When present-ing the study, the authors introduce another model. It becomes unclear what mechanism is implied. The relationship between physical effort and self-control is not clear enough. Some definitions seem necessary to well understand the positioning of the authors.

Second. The labor supply theory needs more explanation. The authors must explain why they consider this model rather than another one. Moreover, the authors have to explain the simi-larities between strenuous physical task and cognitive task. Can we suggest that physical ef-fort and cognitive effort share something in common? The ICWD deserves more explanations.

Third. Methods. There are some repetitions between the general procedure section and the neuromuscular procedure section. Moreover, sections 3.3, 3.4 and 3.5 must be shortened. It is not clear how long the income manipulation task lasts. Moreover, no information is given concerning the instructions given to the participants during the knee extension task. These information are essential to understanding the study.

Fourth. In the discussion section, the explanations are confusing. The authors wrote that the labor supply theory was, for the first time, used to explain self-controlled persistence in a physically demanding task. This cannot be an argument to make this study. The authors have to justify their study by using convincing arguments. Why the labor supply theory is useful in the context of this experiment? What does the model add to the others theories? These ques-tions should be answered in the introduction section. Moreover, the authors have to deepen the discussion section by proposing alternative explanations. For instance, another explanation could be that the time of the depletion task was not long enough to obtain a decrease after the ICWD manipulation. Unfortunately, no information is given concerning the duration of this task. In the implication section, the authors wrote “participants in the ICWD condition increased their time to disengagement”. What does it mean exactly?

Overall, the introduction section is very confusing and the authors have to lighten the method section and deepen the discussion section.

Author Response

Reviewer 3

Review of “Investigating performance in a strenuous physical task from the perspective of self-control” Manuscript Brain Sciences

General comments:

The aim of this study was to test the effect of income compensated wage decrease (considered as ego-depletion task) on persistence in a strenuous physical task. The authors have used the labor supply theory in order to make the hypothesis that self-control could regulate voluntary strenuous physical task. Contrary to prediction, the authors didn’t validate their hypothesis and some explanations are given.

Despite interesting purpose, four main criticisms can be made:

### We thank you for your work and for your comments. We have tried to answer them as best as possible. Overall, we think that these changes have helped the manuscript.

First. the authors wrote that “the capability to endure physical effort is an important feature for species survival”. Can they give some examples to illustrate this point of view? One can be in desagreement with that. The first paragraph of the introduction section is very confusing and must be rewritten. The authors mix several models (cost-benefits, resource) and the understanding becomes complicated. Paragraph 2 is redundant with the first paragraph and deserves clarifications. The authors have to focus on one model and sufficiently describe it to propose clear hypotheses. When present-ing the study, the authors introduce another model. It becomes unclear what mechanism is implied. The relationship between physical effort and self-control is not clear enough. Some definitions seem necessary to well understand the positioning of the authors.

### We have taken into account your comments and have modified the introduction. To avoid the redundancy you have noticed (and rightly so), we have re-written and restructured the first 2 paragraphs. In the first paragraph, we now focus on the notion that self-control is important for sustaining a physically demanding task. In the second paragraph, we then focus on explanations as to why people do not always apply maximum control.

We also agree with you that, in its previous version, the use of multiple models made the paper confusing to read. We have now removed theoretical models that are not strictly necessary and we avoid a conflation of terms in regard to resource or cost-benefit, by focusing squarely on the latter (which is also in keeping with our main model and hypotheses). Finally, as self-control is the core concept of the present research, we now explicitly define it in the first paragraph of the paper.

Second. The labor supply theory needs more explanation. The authors must explain why they consider this model rather than another one. Moreover, the authors have to explain the similarities between strenuous physical task and cognitive task. Can we suggest that physical effort and cognitive effort share something in common? The ICWD deserves more explanations.

### The labor supply theory is an interesting model since it can describe more complex situations than value based binary go/no go models would do. More specifically, it takes into account the choice of a balance between two factors that can be both desired at the same time (e.g. leisure, and the income associated with the work performed) and thus can predict counter-intuitive results. We agree with you that the rationale behind the use of such model was not clear. We have added precisions in the part 2 of the introduction.

Third. Methods. There are some repetitions between the general procedure section and the neuromuscular procedure section. Moreover, sections 3.3, 3.4 and 3.5 must be shortened. It is not clear how long the income manipulation task lasts. Moreover, no information is given concerning the instructions given to the participants during the knee extension task. These information are essential to understanding the study.

### We are sorry but section 3.3 and 3.5 cannot really be reduced. They describe the experiment so it can fully be reproduced and have the standard length and content that are usually seen in papers using these methods. We agree with you that there were some repetitions and that the section was not so clear. We have made changes and tried to make the full methods section clearer. In section 3.4, all instructions given to participants during the task are described. However, we agree with you that we had not included the instructions given before the task. Thanks to your comment, we have now added the most important instructions and have added the full instructions as a supplementary material.

Fourth. In the discussion section, the explanations are confusing. The authors wrote that the labor supply theory was, for the first time, used to explain self-controlled persistence in a physically demanding task. This cannot be an argument to make this study. The authors have to justify their study by using convincing arguments. Why the labor supply theory is useful in the context of this experiment? What does the model add to the others theories? These ques-tions should be answered in the introduction section.

### We agree with you and we thank you for pointing out the lack of rational. We have changed the introduction and tried to answer your questions.

Moreover, the authors have to deepen the discussion section by proposing alternative explanations. For instance, another explanation could be that the time of the depletion task was not long enough to obtain a decrease after the ICWD manipulation. Unfortunately, no information is given concerning the duration of this task.

### We are sorry but the writing in our previously submitted manuscript might not have been clear enough on this! There was no ego depletion task in this study. The changes in behaviour were expected to occur simply because of the changes in the way the wage was delivered (ICWD conditions, as predicted by the labor supply theory). We have tried to make this point clearer in the introduction.

In the implication section, the authors wrote “participants in the ICWD condition increased their time to disengagement”. What does it mean exactly?

### The time to disengagement corresponds to the duration of the knee extension task. This means that the ICWD group had a longer task duration at t2 than at t1. We have rephrased the sentence to remove any confusion: “…participants in the ICWD condition increased task duration at t2”.

Overall, the introduction section is very confusing and the authors have to lighten the method section and deepen the discussion section.

### We thank you for your comments as they have provided us with helpful aspects that we needed to address in the manuscript. We have made substantive changes in the introduction and in the methods part. We could not lighten the methods part too much, but we have tried to make it an easier read. Since we have obtained unpredicted results, their discussion remains quite speculative. In this situation, we feel that adding more potential explanations without the support of more data may not benefit the overall manuscript (e.g. ego-depletion explanations). However, if the reviewer/editor has the impression that this part needs to be extended, we can of course do this.